# Acquired Mechanisms of Resistance to Osimertinib—The Next Challenge

**DOI:** 10.3390/cancers14081931

**Published:** 2022-04-12

**Authors:** Alejandro Ríos-Hoyo, Laura Moliner, Edurne Arriola

**Affiliations:** 1Department of Medical Oncology, Hospital del Mar-CIBERONC (Centro de Investigación Biomédica en Red de Oncología), 08003 Barcelona, Spain; arios@psmar.cat (A.R.-H.); laura.molinerjimenez@nhs.net (L.M.); 2Cancer Research Program, IMIM (Institut Hospital del Mar d’Investigacions Mèdiques), 08003 Barcelona, Spain

**Keywords:** osimertinib, NSCLC, EGFR, mechanisms of resistance

## Abstract

**Simple Summary:**

Osimertinib has revolutionized the treatment of EGFR-mutated tumors. Its current applications include the first-line setting, second-line setting, as well as the adjuvant setting. Although it represents a milestone in the context of targeted therapy, inevitably all tumors develop an acquired resistance, some mechanisms involve EGFR, others do so through alternative pathways leading to a bypass in osimertinib inhibition. It is key to understand these acquired mechanisms of resistance, both in the clinical setting, as well as in preclinical models, in order to develop and contribute to the identification of possible therapeutic strategies to overcome this acquired resistance.

**Abstract:**

EGFR-mutated tumors represent a significant percentage of non-small cell lung cancer. Despite the increasing use of osimertinib, a treatment that has demonstrated an outstanding clinical benefit with a tolerable toxicity profile, EGFR tumors eventually acquire mechanisms of resistance. In the last years, multiple mechanisms of resistance have been identified; however, after progressing on osimertinib, treatment options remain bleak. In this review, we cover the most frequent alterations and potential therapeutic strategies to overcome them.

## 1. Introduction

Non-small cell lung cancer (NSCLC) accounts for approximately 80–90% of lung cancers; adenocarcinoma represents the most frequent histologic subtype of NSCLC. Different molecular oncogene addiction drivers have been identified in NSCLC, particularly in adenocarcinomas, including *EGFR* mutations, *ALK* and *ROS1* rearrangements, *MET* mutations and amplifications, *BRAF* and *KRAS* mutations, *RET* fusions, *NTRK* fusions, and *HER2* mutations, among others [1,2]. *EGFR* mutations were first described in 2004. The most frequent mutations are exon 21 point L858R mutation and in-frame deletions of exon 19, accounting for 40% and 50% of *EGFR* mutations, respectively, and conferring sensitivity to EGFR inhibitors. Other described mutations include G718X, S768I, and L861Q, as well as an in-frame insertion in exon 20 [3,4,5]. In NSCLC, *EGFR* mutations display a heterogeneous representation depending on ethnicity and region, accounting for 40–60% in South-East Asian patients, 9–67% in patients from the Americas, 22–27% in patients from the Indian subcontinent, around 12% in African Americans, and 10–20% in Caucasian patients [4,6,7]. First-generation EGFR inhibitors include erlotinib, gefitinib, and icotinib, which reversibly bind to EGFR and inhibit the binding of ATP to the tyrosine kinase domain. The second-generation EGFR inhibitors, afatinib and dacomitinib, bind covalently to EGFR, thus irreversibly inhibiting its activity [8]. After an initial benefit with the use of EGFR inhibitors, acquired resistance invariably develops; approximately 60% of the patients develop the T790M mutation, and other mechanisms of resistance include *EGFR* amplification concurrent with T790M, *HER2* amplification, *MET* amplification, SCLC transformation, and others [9]. Third-generation EGFR inhibitors include osimertinib, rociletinib, and olmutinib; however, only the first is approved in current clinical practice. Osimertinib binds covalently to the C797 residue in the ATP-binding site of mutant *EGFR* [10].

Osimertinib was designed to overcome the effects of the T790M EGFR-resistant mutation while targeting the initial *EGFR* activating mutation [11]. It was first tested in the second-line setting, showing an increase in median progression-free survival (mPFS) compared to platinum plus pemetrexed based chemotherapy in those patients whose T790M-positive tumors had progressed to an EGFR-TKI therapy [12]. Subsequently, the results of the FLAURA clinical trial, demonstrating improvements in PFS and OS compared to first-generation EGFR TKIs, led to osimertinib being approved as frontline treatment [13,14].

Despite all these achievements, significant challenges remain ahead. Patients with *EGFR* mutated tumors seem to benefit less, if anything, from immunotherapy compared to other patients with advanced NSCLC and no targetable alterations. Clinical trials combining osimertinib and immunotherapy showed intolerable toxicity, and, on the other hand, most clinical trials combining chemo- and immunotherapy lacked a significant number of patients with *EGFR* mutant NSCLC to draw definitive conclusions on efficacy [15].

It is now more essential than ever that we understand the mechanisms of resistance of NSCLC *EGFR* mutant tumors. In this review, we present and analyze the most frequent alterations associated with resistance to osimertinib (Figure 1), potential therapeutic strategies in this scenario (Table 1), in vitro studies assessing mechanisms of resistance to osimertinib (Table 2), and potential therapeutic strategies.

To identify published articles or abstracts that described acquired mechanisms of resistance against first- or second-line osimertinib in clinical practice, as well as in vivo and in vitro models, we systematically searched PubMed (https://pubmed.ncbi.nlm.nih.gov/, accessed on 1 July 2021) for relevant studies. Our search criteria included the following terms: “osimertinib”, “resistance”, “non-small cell lung cancer”, “clinical trials”, “cell lines”, as well as keywords related to acquired mechanisms of resistance, including but not limited to “C797X”, “small-cell transformation”, “squamous cell transformation”, “MET amplification”, “HER2 amplification”, “HER2 mutation”, “RET fusions”, “BRAF mutations”, “BRAF amplification”, and “RAS mutations”.

## 2. EGFR-Dependent Mechanisms of Resistance

### 2.1. C797X Mutations

Osimertinib selectively blocks mutated-EGFR by irreversibly binding to its C797 residue. As expected, the most common EGFR-dependent mechanisms of resistance are based on mutations in this spot, usually a substitution to serine, leading to the C797S mutation [63].

In vitro studies evaluating the EGFR C797S mutations have mostly been achieved through the induction of genetic alterations in cell lines and not through exposure to osimertinib. For instance, NIH3T3 cells (an immortalized mouse embryonic fibroblast cell line) were used and transduced with lentiviruses bearing genes encoding the del19/T790M/C797S and L858R/T790M/C797S mutations in order to study the growth inhibitor effects of the compound CH7233163 [25]. Another mechanism to induce the C797S mutation was used in Ba/F3 cells (a murine, IL-3 dependent, hematopoietic cell line) harboring del19/L858R +/− T790M through a site-directed mutagenesis kit, developing constructs which were later transferred to the retroviral vector JP1540 or lentiviral vector JP1698 in cells infected with the viruses. after exposure to the EGFR TKI WZ4002 for two weeks, resistant clones were selected, identifying the previously transduced C797S [55,64].

The incidence of C797X in the clinical setting differs depending on the treatment setting of osimertinib. In the first-line scenario, 6 of 91 patients (7%) acquired a C797X mutation in plasma samples from the FLAURA clinical trial [65]. Analysis of ctDNA from plasma samples from 73 patients who progressed to osimertinib in the second line in the AURA3 trial showed the emergence of C797X mutations in up to 15% of the patients, 91% of them corresponding to C797S [66]. In the second-line setting, Oxnard et al. found C797S in 22% of 41 patients from tissue biopsies at progression [67]. Interestingly, the T790M mutations consistently remained detectable in all patients with a C797X mutation.

Several therapeutic strategies have been proposed to tackle the emergence of these mutations; however, currently, chemotherapy remains the standard treatment after progression to osimertinib.

Preclinical data suggest that, in the absence of T790M, cells remain sensitive to reversible binding EGFR TKIs (gefitinib, erlotinib), whose binding does not rely on C797 [68]. However, the only clinical information regarding this strategy is based on isolated clinical cases. One clinical case treated a patient after progression to first-line osimertinib with erlotinib, achieving a response that lasted 5 months [17].

In the second-line setting, the position in which T790M is located, whether in the same (cis) or in a different (trans) position related to C797S, has a significant impact. In vitro studies have reported that trans-T790M/C797S cells were resistant to third-generation EGFR TKIs but sensitive to the combination of first- and third-generation TKIs, whereas cis-T790M/C797S cells were refractory to all EGFR TKIs tested, as well as their combinations [68].

Case reports explored this clinical approach: a patient with a triple mutation tumor (del19/trans-T790M/C797S) received osimertinib and erlotinib after progression to osimertinib, reaching a PFS of 3 months [69]. In another reported case, a patient with a similar clinical scenario received osimertinib, erlotinib, and bevacizumab, which resulted in an 8-months PFS [18].

In the cis-T790M/C797S population, in vitro studies suggest that brigatinib (an ALK and ROS1 inhibitor with EGFR inhibition activity) could overcome this resistance mechanism with the combination of brigatinib plus an anti-EGFR antibody [70]. A retrospective cohort of patients who progressed to osimertinib with cis-T790M/C797S, treated with a combination of brigatinib and cetuximab, obtained an ORR (overall response rate) of 60%, a DCR (disease control rate) of 100%, and an mPFS (median progression-free survival) of 14 months [19].

Preliminary data was presented recently about patritumab deruxtecan (HER3-DXd), an antibody–drug conjugate (ADC) against HER3 [20]. Forty-four patients with *EGFR* mutant NSCLC who had received osimertinib and had a median of four prior lines of therapy obtained an ORR of 39% (with complete responses in 2% of the patients) and an mPFS of 8.2 months. The efficacy of patritumab deruxtecan was maintained across various resistance mechanisms and different levels of HER3 expression. Interestingly, in the subgroup of patients with a known *EGFR* resistance mutation, ORR rose to 50%. This promising drug will be further evaluated in the HERTHENA-Lung01 trial [71] and also in combination with osimertinib [72].

Preventive strategies are also being tested in patients with *EGFR*-mutated untreated NSCLC. One phase I/II study is exploring whether the combination of osimertinib and gefitinib could delay the emergence of acquired resistance mechanisms. To date, the combination has shown an acceptable toxicity profile [73].

### 2.2. Less Common EGFR-Dependent Alterations

Less frequent mutations different to C797X have also been described. After progression to osimertinib in the FLAURA study [65], 2% of patients acquired the L718Q and 1% acquired the S768L mutations, respectively. In the AURA3 study [66], L792X emerged in 3% of the patients who progressed; G796X, L718Q, and exon 20 insertions were described at a frequency of 1% each. In a cohort of 93 Asian patients who received osimertinib as second-line treatment, G796X, G719A, L718Q, and L792X emerged in 2%, 2%, 8%, and 12% of cases, respectively [74]. Most of these alterations occur concurrently with other mutations, which suggests the appearance of clonal heterogeneity at progression. L718Q is a notable exception, as it usually emerges isolated.

There is limited evidence regarding the functional effect of these less frequent mutations. G796X mutations occur close to C797 residue and thus, affect the interaction between osimertinib and its binding residue [75]. L792X, L718Q, and G719X mutations also decrease the interaction between osimertinib and its binding residue [76,77].

A case report by Fang et al. described a patient whose *EGFR* L858R/L781Q tumor responded partially to a third-line of treatment with afatinib before progressing and identifying a *KRAS* mutation [78]. For L792X mutations, in vitro studies using different EGFR inhibitors and combinations have failed to show any sign of response [77]. To our knowledge, other strategies to address these rare mutations are not published.

Another *EGFR*-dependent mechanism of resistance is the amplification of *EGFR*, with an incidence of 33% and 31% in the first and later lines, respectively. In vitro studies have confirmed *EGFR* amplification as an independent resistance mechanism to osimertinib, even though, in most cases (up to 50%), it co-occurs with other *EGFR* resistance mutations [62,74,79]. One branch of the ORCHARD trial is currently evaluating the combination of osimertinib and necitumumab in this population [80].

### 2.3. Fourth-Generation EGFR TKIs as a Strategy to Overcome Resistance to Osimertinib

Given that the C797S mutation is the most frequently acquired resistance mechanism to osimertinib, fourth-generation EGFR TKIs are currently being developed. We describe some of the most relevant in the following section.

#### 2.3.1. EAI045

EAI045 is the first allosteric reversible, non-ATP competitive inhibitor targeting L858R/T790M/C979S *EGFR* co-occurring mutations [21]. In vitro and in vivo studies confirmed that EAI045 and cetuximab were effective in inhibiting L858R/T790M/C979S cells, as well as in mice models with L858R/T790M, leading to an inhibition of the downstream signaling proteins. It is noteworthy that EAI045 was not effective in inhibiting exon19del/T790M mutation models [22]. EAI045 has also shown activity inhibiting the exon 21 L861Q mutation, as reported in preclinical studies [23]. No clinical trials were found studying this molecule.

#### 2.3.2. JBJ-04-125-02

JBJ-04-125-02 is a reversible, non-ATP competitive allosteric inhibitor that was effective in preclinical models harboring the L858R/T790M/C979S mutations in combination with osimertinib, leading to increased apoptosis and an inhibition of cellular growth. JBJ-04-125-02 lacks binding affinity against exon19del; therefore, it is ineffective in this situation [24]. At the moment, we could not find any clinical trials evaluating this drug.

#### 2.3.3. CH7233163

CH7233163 is a non-covalent ATP-competitive inhibitor, which has shown in vitro and in vivo activity overcoming the *EGFR* del19/T790M/C797S, L858R/T790M/C797S, del19/T790M, L858R/T790M, del19, and L858R mutations [25]. However, no clinical trials testing this molecule are ongoing at the moment. Nonetheless, CH7233163 appears to be an attractive drug to be tested in this setting.

#### 2.3.4. BLU-945

BLU-945 is a potent, selective EGFR inhibitor with activity against del19/T790M/C797S, L858R/T790M/C797S, del19/T790M, and L858R/T790M in in vitro and in vivo assays [26,27]. The phase I clinical trial SYMPHONY is currently evaluating the use of BLU-945 in the second-line for patients whose tumors acquired C797S [81].

#### 2.3.5. Other Drugs

Early evidence is available for other inhibitors. TQB3804 is a drug that effectively inhibited del19/T790M/C797S, L858R/T790M/C797S, del19/T790M, and L858R/T790M in in vitro and in vivo assays [82]. TRE-069 has shown preclinical data as an EGFR del19/T790M/C797S inhibitor [83].

## 3. Histologic and Phenotypic Transformation

### 3.1. Small Cell Transformation

The underlying mechanisms that cause a number of *EGFR* mutant tumors to transform into small cell lung cancer (SCLC) are unknown; limited data due to the lack of tissue availability makes it difficult to understand this phenomenon [84]. Its incidence has been reported between 6% and 15% in the second-line setting, being less common in the first-line setting, with up to 4% of cases reported [79,85,86].

Tissue biopsy at the time of progression to osimertinib remains of paramount importance in order to identify this transformation; however, in several cases, elevated serum levels of neuron-specific enolase can be found and suggest this phenomenon [87].

Consistently, most of the transformed tumors maintain the original activating *EGFR* mutation [88]. However, after transformation, *EGFR* expression levels drop, which explains why these tumors are not sensitive to EGFR-TKI therapy [89]. Common mutations identified in samples following transformation are *TP53* (91%), *Rb1* (58%), and PIK3CA (27%). Interestingly, in this study, these mutations were not described prior to histologic transformation [28].

Among the population of *EGFR*-mutated advanced NSCLC, a higher risk population for this transformation into SCLC has been defined by the presence of the concurrent loss of both *TP53* and *RB1*. This population represents approximately 5% of all *EGFR* mutated tumors, and its incidence of transformation is 18%, in contrast to *EGFR* mutant tumors without mutation on *RB1* and *TP53*, representing 3% of the cases. These patients have worse outcomes, with median OS in a retrospective cohort of 29.1 months compared to 56.4 months of patients with preserved *RB1* and *TP53* [90]. A retrospective study identified that pretreatment *Rb1* loss was significantly associated with SCLC transformation. The study also identified patients with worse outcomes as those with concurrent pretreatment *TP53* and *Rb1* loss [91].

A retrospective analysis examined a cohort of 67 patients diagnosed with transformed-SCLC, where up to 30% had received osimertinib. The most common treatment was platinum-etoposide, achieving an ORR of 54%, mPFS of 3.4 months, and mOS of 10.9 months since the transformation [28]. Interestingly, there were no responses among the 17 patients who were treated with nivolumab in monotherapy or combination with ipilimumab. To date, platinum-etoposide chemotherapy is the only treatment strategy with confirmed clinical efficacy. An ongoing clinical trial is testing the combination of osimertinib and platinum-etoposide chemotherapy [29]; the role of chemo-immunotherapy combinations remains unknown in this population.

### 3.2. Squamous Cell Transformation

Schoenfeld et al. identified five cases of transformation into squamous cell carcinoma [79]. The incidence was similar in first-line osimertinib, with 7%, and later-lines, with 9%. All squamous cell-transformed tumors maintained the original *EGFR* mutation. There was no clear molecular pattern after evolving into squamous tumors, with only one patient gaining a PIK3CA mutation.

Currently, there are no clinical data regarding therapeutic strategies in these patients; however, a histology-based approach is recommended [30].

## 4. *MET* Amplification

The *MET* (mesenchymal-epithelial transition factor) gene is located on chromosome 7q21-q31. It encodes for the MET polypeptide, which is processed into a glycoprotein and serves as a transmembrane receptor tyrosine kinase. MET becomes activated through the binding of the hepatocyte growth factor ligands, subsequently activating different pathways, such as RAS/ERK/MAP kinase, PI3K/AKT, Wnt/β-catenin, and STAT [92,93].

*MET* gene amplification has been identified as a resistance mechanism against osimertinib in HCC827 cells, which is a hypersensitive *EGFR* exon 19 mutant NSCLC cell line [94]. Induction of resistance to osimertinib was achieved by exposing cells to gradually increasing concentrations of osimertinib (initially at 10 nM up to 500 nM) for 6 months. *MET* amplification, as well as hyperactivation, were detected [56] in the osimertinib resistant cells.

In clinical practice, *MET* amplifications are found between 9% and 24% of tumors that progress to osimertinib as second or later lines [66,67,79,85]. After progression to first-line osimertinib, *MET* amplification was found between 7% and 15%, representing the second most common resistance mechanism after C797X [65,79].

Multiple clinical trials are currently open to find effective strategies in this setting of unmet need, especially since osimertinib is now the current standard of treatment in the first line. Different combinations of both anti-EGFR and anti-MET drugs show promising results for these patients.

The TATTON phase 1b study assessed the combination of osimertinib and savolitinib in patients with *EGFR*-mutated and *MET*-amplified tumors across different clinical scenarios [31]. The cohort B1 consisted of 69 patients who had previously received a third-generation TKI, achieving an ORR of 30%, a DCR of 75%, and an mPFS of 5.4 months. Even though responses were considerably lower than in other cohorts, it represents a significant achievement considering that up to one-third of the patients received over three lines of treatment. Based on these results, as well as on its safety profile, an additional phase 2 trial was initiated [95]. One of the cohorts of the multi-arm clinical trial ORCHARD also examines this combination [19].

The phase I clinical trial CHRYSALIS evaluated the combination of lazertinib (a third-generation EGFR TKI) and amivantamab (a bi-specific antibody that targets both MET and EGFR) in patients with osimertinib-relapsed and chemotherapy-naïve advanced NSCLC [32,96]. In this study, patients with an identified *MET* or *EGFR/MET*-based mechanism of resistance had a 50% ORR. Interestingly, the response rate rose to 90% in 10 patients with high expression by immunohistochemistry (defined as a combined EGFR and MET H-score > 400). However, of these high expressors, five of them had an unknown mechanism of resistance. The results of the general cohort showed an ORR of 36%, with an mDoR (median Duration of Response) of 9.6 months and an mPFS of 4.9 months. A cohort of the future CHRYSALIS-2 trial will try to validate these findings.

The INSIGHT 2 trial is evaluating tepotinib (a MET TKI) in monotherapy and its combination with osimertinib in patients who progressed to first-line osimertinib with an acquired *MET* amplification [33].

Several case reports have used the combination of osimertinib and crizotinib in patients whose tumors had progressed to osimertinib as a second or later line and had acquired a MET amplification. Responses in the published cases differ, with benefits ranging from 2 to 7 months [34,35,97].

Responses were also seen in the clinical trial that evaluated patritumab-deruxtecan; however, only 8% of the tumors had *MET* alterations in the pretreatment samples. Consequently, efficacy data in this setting is currently insufficient to draw any conclusion [98].

## 5. *HER2* Alterations

HER2 is a tyrosine kinase receptor that belongs to the EGFR family. Among the different HER family proteins, HER2 has the strongest catalytic kinase activity [57,99]. Its phosphorylation leads to the downstream activation of the PI3-Akt, MAPK, and ERK MET/MAPK pathways [1,2,3]. Aberrations in HER2 can be found in NSCLC, including amplifications and mutations, both leading to HER2 activation [6].

In vitro studies have developed the HER2 exon 16 skipping (HER2D16) in HEK293 cells (human embryonic kidney cell line) [100] and H1975 cells (T790M/L858R) through plasmid transfection. In HEK293, HER2D16 was able to induce the phosphorylation of ERK, confirming the signaling activity of this mutation. H1975 cells with HER2D16 were confirmed to be resistant to osimertinib, pointing out that the HER2 interaction with EGFR was able to form heterodimers, thus maintaining the phosphorylation of EGFR, as well as AKT and ERK [57].

In the clinical setting, *HER2* aberrations have been identified as acquired mechanisms of resistance to osimertinib. In the first-line therapy, amplification was detected in 2% of the cases and *HER2* mutations in 1% [65]. In second-line osimertinib, findings include *HER2* amplifications (5%), *HER2* amplifications co-occurring with EGFR L792X + C797X + PIK3CA amplification (1%), *HER2* amplifications and EGFR G796S + MET amplification (1%), and *HER2* amplifications and PIK3CA amplifications (1%) [66]. The most common mutations reported include in-frame exon 20 insertions, as well as exon 16 skipping *HER2* deletion [57,101,102]. Preclinical models have shown the efficacy of osimertinib in *HER2* amplification [16,101], however, these results have not been translated to a clinical benefit.

In the clinical trial evaluating patritumab-deruxtecan, some patients presented with *HER2* alterations (mutations or amplifications) and benefited from the drug in terms of response [20].

Different targeted therapies against HER2 have been tested in NSCLC without co-occurring *EGFR* mutations, including lapatinib, neratinib (HER2 and EGFR inhibitors) [36,37,103], and trastuzumab-deruxtecan (T-DXd, an ADC against HER2, linked to a topoisomerase I inhibitor). These combinations would represent an attractive strategy to overcome this resistance mechanism.

## 6. *RET* Alterations

The *RET* proto-oncogene encodes a receptor tyrosine kinase (RTK). Its activation can develop as a consequence of gain of function amino acid substitutions and genomic rearrangements, leading to the formation of fusion proteins; *RET* fusions are usually generated by pericentric and paracentric inversions of chromosome 10 [104,105]. *RET* alterations frequently coexist with other genomic alterations such as TP53, cell cycle-associated genes, the PI3K pathway, and mitogen-activated protein (MAP) kinase effectors [104]. In the AURA3 study, RET-ERC1 fusions were reported in 1% of the cases as a mechanism of resistance to osimertinib [66]. Other RET fusions include CCD6-RET, NCOA4-RET, and MYH9-RET [85,106,107]. Rearrangements have also been described in the *RET* region of exon 11 to intron 11, which are sites of fragile DNA secondary structures [108]. Liquid biopsy is useful in detecting tumor heterogeneity, particularly when diverse resistance mechanisms develop in different tumor sites, as reported in the case of an NSCLC with an EGFR exon 19del and T790M, which had a liver progression with an acquired CCDC6-RET fusion. The patient received treatment with selpercatinib and responded to this combination [39].

## 7. *BRAF* Alterations

BRAF is a member of the RAF family of serine/threonine kinases. It is a part of the MAP kinase pathway, and its signaling follows downstream from BRAF to MEK 1 and 2 and ERK, which further phosphorylates multiple molecules [109].

In vitro studies have identified mutations in *BRAF* G469A as a resistance mechanism against osimertinib in PC9 cells (*EGFR* exon 19 delE746-A750). The cells were exposed to increasing concentrations of osimertinib (ranging from 10 to 500 nM) for nine months. When identifying possible resistance mechanisms, *EGFR*, *HER2*, or *MET* alterations were not identified, however, a BRAF G469 mutation was detected. The study observed that this mutation maintained the activity of the MAP kinases pathway. Combination treatment with osimertinib plus selumetinib or trametinib (both MEK 1/2 inhibitors) was effective in restoring the sensitivity of osimertinib to resistant BRAF G469A mutated cells [42].

In the clinical trial AURA3, *BRAF* alterations were reported as a mechanism of resistance to osimertinib in 3% of the cases (BRAF V600E mutation). This mutation co-occurred with *MET* amplification and FGFR3-TACC3 fusion, with concurrent EGFR C797X mutation; other co-occurring alterations included *MET* amplification with BRAF V600E and *CDK6* amplification [66,109,110]. Analysis of the FLAURA study detected BRAF V600E in 3% of the cases as a mechanism of resistance to first-line osimertinib [65]. A combination of osimertinib with BRAF and MEK inhibitors, dabrafenib, and trametinib has been used effectively in the case of BRAF V600E acquired mutation and *EGFR* exon 19del/T790M, which had progressed to osimertinib, with acceptable tolerance to the treatment [111]. The use of osimertinib in combination with a single BRAF inhibitor, vemurafenib, has been reported as a successful strategy in overcoming BRAF V600E acquired resistance to osimertinib [41].

*BRAF* fusions represent approximately 2% of the cases of acquired resistance to osimertinib; described fusions include *PJA2-BRAF*, *MKRN1-BRAF*, and *AGK-BRAF*. The latter was shown to develop in the primary tumor but was absent in a metastatic lesion, highlighting the impact of clonal heterogeneity in resistance mechanisms [112,113]. The use of osimertinib in combination with the MEK inhibitor trametinib has been reported as a fifth line therapy in a patient with NSCLC and *EGFR* exon 19del and an *AGK-BRAF* fusion, observing a partial response to this treatment [114].

## 8. *KRAS* Mutations

KRAS is a member of the membrane-bound family proteins RAS. It possesses inherent GTPase activity. RAS can activate different effector molecules, such as RAF and the MAP kinase pathway, as well as PI3K, ultimately activating mTOR [115].

In vitro studies have detected RAS alterations as a resistance mechanism against first-line osimertinib in PC9 cells, developing an *NRAS* E63K mutation. As well as *NRAS* G12V and G12R, *KRAS* copy number gains were also detected. PC9 cells were chronically treated with escalating concentrations of osimertinib and a single concentration of osimertinib; different models were used, exposing cells to a final concentration of osimertinib of 160 nM and 1500 nM in the different models. The treatment combination of osimertinib with either selumetinib or an Aurora kinase b inhibitor was effective in overcoming resistance to osimertinib [43].

In the clinical setting, *KRAS* mutations were reported as acquired resistance mechanisms in 1% (*KRAS* G12D) and 3% (*KRAS* A1467T, *KRAS* G12C, and *KRAS* G12D, 1% each mutation) in the AURA3 and FLAURA trials, respectively [65,66]. Other reported acquired resistance mechanisms include co-existing alterations, such as the loss of T790M and development of C797S, with different *KRAS* mutations, including G12D, G12S, G61K, and Q61R, as well as an amplification of *CDK4/KRAS/MDM2* [67,116,117,118,119]. To the best of our knowledge, no strategies have been developed to overcome this acquired resistance mechanism; however, the concomitant use of osimertinib with novel KRAS G12C inhibitors, such as sotorasib [44] or adagrasib [45], could be an attractive alternative in this particular mutation.

## 9. *PI3K* Alterations

PI3K is activated through different upstream pathways involving tyrosine kinases, G coupled proteins, and RAS-related GTPases. Its activation can lead to diverging downstream pathways, such as Akt, TEC family tyrosine kinases, and mTOR [120]. PIK3CA mutations or amplifications were reported as an acquired resistance mechanism in the AURA3 trial, presenting, in coexistence with *HER2* amplification, *CCND2* and *CCNE1* amplifications. PIK3CA E545K was the most frequent mutation detected [66]. In the FLAURA trial, PIK3CA mutations were detected in 6% of the cases (E545K 4%, E453K 1%, H1047R 1%) [65]. An in vitro study observed that the PIK3CA H1047R mutation drives resistance to osimertinib. Co-treatment of cells with osimertinib and alpelisib, a PIK3CA inhibitor, resulted in a downregulation of the AKT signaling pathway [48]. A study involving 605 patients with NSCLC detected up to 14.9% PIK3CA, PTEN, or Akt mutations in patients who had progressed to EGFR inhibitors. Subsequently, six patients were treated with EGFR TKIs and everolimus, an mTOR inhibitor. This combination resulted in a limited antitumoral activity with stable disease in five patients and a progressive disease in one [46,47].

## 10. Cell Cycle Gene Alterations

Cyclin D-dependent kinases (CDK4 and CDK6) are major oncogenic drivers; their sustained activation leads to cancer cells entering the cell cycle repeatedly by producing G1-S phase transitions and reducing the duration of the G1 phase. Genes associated with CDKs include *CCND*, *CCNE*, and *CDKN*, among others [121].

Cell cycle alterations have been described in in vitro studies as a resistance mechanism to osimertinib in H1975 cells (*EGFR* L858R and T790M mutations) through an increased expression of CDK4. Osimertinib resistance was achieved by exposing the cells to osimertinib concentrations from 5 nM to 1.5 µM for 22 weeks. Subsequently, H1975 OR (osimertinib resistant) cells were developed. No alterations in *MET*, *KRAS*, *BRAF*, *MEK*, or *PI3K* were detected, and neither C797X nor T790M loss was identified. H1975 OR cells had fewer G1 phase and more G2 phase cells than H1975 cells [49]. A combination of palbociclib, a CKD4/6 inhibitor, with osimertinib in an in vitro study was effective in controlling tumor cells proliferation [49]. Similar results were observed in an in vitro model, where a combination of abemaciclib, a CDK 4/6 inhibitor, and osimertinib inhibited the onset of resistance to osimertinib [50].

In the AURA3 trial, cell cycle gene alterations were acquired in 12% of samples, as follows, mutation of CDKN2A E27fs 1%, and amplifications of *CCND1* 1%, *CCND2* 1%, *CCNE1* 7%, and *CDK6* 7% [66]. In the FLAURA study, the alterations reported were amplifications of the following genes: *CCND1* 2%, *CCND2* 1%, *CCND3* 1%, *CCNE1* 2%, *CDK4* 2%, and *CDK6* 3% [65]. Other studies have reported cell cycle alterations in up to 26.3% of cases after progression to osimertinib [122]. These genomic alterations have been associated with poor outcomes regarding progression-free survival and overall survival in the context of osimertinib treatment [123]. A clinical trial with osimertinib in combination with a CDK 4/6 inhibitor in a population with *EGFR* mutated NSCLC with or without T790M is evaluating the efficacy of this combination; however, this trial excludes patients who previously received osimertinib [124].

## 11. AXL Overexpression

AXL is a receptor tyrosine kinase, which belongs to the tumor-associated macrophage family (TAM), including TYRO-3 and MER. AXL ligand is Gas6, which binds to the ectodomain of AXL; its activation leads to cellular growth, proliferation, motility, and invasion, involving different signaling pathways, such as MAP kinase and PI3K/Akt, among others. Furthermore, AXL has been implicated in the process of epithelial-mesenchymal transition [125,126].

In vitro studies have identified a concurrent *MET* amplification, as well as an AXL upregulation, as a mechanism of acquired resistance in EFGR TKI resistant NSCLC cell lines [127,128]. Furthermore, AXL induced reactivation of HER3, MET, and EGFR was associated with maintaining cell survival and resistance to osimertinib [129]. A study in cell lines evaluated the use of cabozantinib and osimertinib in osimertinib-resistant NSCLC with AXL upregulation, observing a significant tumor suppression [128].

AXL activation has been described as an acquired resistance mechanism to first-line osimertinib in HCC827 cells (*EGFR* exon 19 delE746-A750), PC9 cells, H1975 cells, and HCC4006 cells (*EGFR* exon 19 delL747-A750, P ins) [58,59,60]. HCC827 cells were treated with 30 nM of osimertinib for 3 days and then cultured in a drug-free medium for more than 6 months, subsequently generating osimertinib-resistant cells with *MET* amplification and AXL upregulation identified by Sanger sequencing [127]. In another study, HCC827, HCC4006, PC-9, HCC4011, and H1975 cells were used to develop acquired osimertinib-resistant cell lines through exposure to osimertinib with an escalation method ranging from 10 nmol/L to 2 µmol/L over 6 months, or through a high concentration method, exposing cells to osimertinib at a 2 µmol/L over 6 months. *MET* amplification and AXL expression were detected; however, T790M, C797S mutations of *KRAS*, *NRAS*, *BRAF*, and *TP53* mutations were not detected [128].

AXL overexpression has been reported as an acquired mechanism of resistance to first- and second-generation EGFR TKIs, as well as to second-line osimertinib [126,130]. In NSCLC EGFR mutated patients, baseline AXL overexpression was associated with a decreased response to first-line osimertinib, compared to non-overexpressing tumors [131].

Enapotamab vedotin and ADC specific against AXL has shown activity in an in vivo model of osimertinib resistant NSCLC [51] and is currently being tested in a Phase 1/2 clinical trial including different tumors [52].

## 12. Insulin-like Growth Factor (IGF)-1 Receptor Activation

The IGF-1 receptor belongs to the insulin receptor family and has roles in cell growth and differentiation. It can be activated by IGF1, IGF2, and insulin [132].

Activation of the IGF-1 receptor as an acquired resistance mechanism to osimertinib was described in in vitro studies using PC9 and H1975 cells. PC9 cells were exposed to gefitinib over 6 months, developing resistance through the T790M mutation. Subsequently, these cells were cultured with stepwise escalation to osimertinib in concentrations ranging from 150 nmol/L to 1 µmol/L over 6 months, thus developing osimertinib resistant cells. H1975 cells were exposed to osimertinib using the high-concentration method, culturing cells with 1µmol/L osimertinib for 3 months. Whole exome sequencing of several genes, including *EGFR*, *MET*, *KRAS*, *MEK*, *BRAF*, and *PIK3CA*, was performed. No alterations were detected; however, the IGF-1 receptor was detected to be overactivated in these cells [53].

Activation of the IGF-1 receptor through IGF-2 overexpression has been identified both in clinical specimens and in cell lines of NSCLC *EGFR* mutated osimertinib resistant and proposed as a mechanism of acquired resistance. Furthermore, treatment with linsitinib (an IGF-1 receptor inhibitor) and osimertinib restored sensitivity to osimertinib in in vitro studies [53,133]. We could not find any clinical trials evaluating the combination of linsitinib with osimertinib or other TKIs to overcome this resistance mechanism.

## 13. Epithelial-Mesenchymal Transition (EMT)

EMT is a cellular process involved in different types of cancer, allowing cells to enhance invasive capacity, cancer stem-cell similar properties, as well as resistance to treatments. In this process, epithelial cells lose their cell polarity and cell-to-cell adhesion, including a downregulation of epithelial proteins, such as E-cadherin, and acquire mesenchymal characteristics, including increased migration and invasion properties, and an upregulation of proteins, such as N-cadherin and vimentin. EMT develops through the involvement of different proteins and pathways, including TGF-β, SMAD and MAP kinase pathways, induction of IGF-1 receptor, and Notch signaling, among others [134,135,136].

In vitro studies using H1975/AR cells (gefitinib resistant) were exposed to osimertinib through a stepwise escalation process, later detecting a decreased E-cadherin and increased vimentin expression, as well as the capability to form larger spheroids [61]. In a different study with H1975 OR cells, mRNA expression was evaluated, detecting an upregulation of N-cadherin and vimentin and a downregulation of E-cadherin, which was further confirmed by a Western-Blot analysis [137], supporting EMT as an additional mechanism of resistance to osimertinib.

## 14. Other Rare Acquired Resistance Mechanisms

Other mechanisms of resistance to osimertinib have been described in in vitro studies, including the Src-AKT pathway and EGFR wild-type amplification in PC9 and H1975 cells. The cells were cultured with osimertinib at an initial concentration of 0.3 µmol/L and incrementally increased to 1 µmol/L. After several months of exposure, osimertinib resistant cells were developed [62].

Other reported resistance mechanisms in the clinical setting to first and second-line osimertinib include fusions of SPTBN1-ALK, EML4-ALK, FGFR3-TACC3, NTRK1-TMP3, and GOPC-ROS1, among others [65,66,138,139].

## 15. Conclusions

Osimertinib has become the standard of care in the first-line setting for patients with advanced lung cancer and sensitizing *EGFR* mutations. Despite the long duration of response with osimertinib, eventually, all patients will progress. Up until now, chemotherapy remains the standard treatment in this setting. However, in the advent of new diagnostic genomic platforms, a myriad of diverse mechanisms of resistance to osimertinib have been identified. Different molecular mechanisms can coexist, as well as histologic transformation and molecular derangements. It is therefore of paramount importance to assess these genomic and phenotypic changes through a rebiopsy when feasible and alternatively using liquid biopsies. One of the main challenges when multiple alterations are found is to elucidate which mechanism is driving the resistance to efficiently target that mechanism and improve the chances of therapeutic success. In vitro studies can contribute to this. Another important challenge is that the mechanisms of resistance are diverse, thus, making it complex to design clinical trials encompassing all possible scenarios. Umbrella trials such as ORCHARD are good examples of maximizing the efficiency to address different mechanisms of resistance in one trial.

Another strategy is designing drugs that can potentially overcome different mechanisms of resistance, such as patritumab-deruxtecan or even resistance mechanisms to different drivers, e.g., the phase 1 TROPION-PanTumor01 study. In this trial, datopotamab deruxtecan, an anti-TROP2 ADC, was used to treat patients with NSCLC and targetable genomic alterations (including EGFR mutation and ALK and RET fusions) who had progressed to prior treatment options (82% of patients had received ≥3 lines). The ORR was 35%, and the median DoR was 9.5 months, thus, positioning datopotamab deruxtecan as an attractive option in the future [105]. The outcomes of larger phase 2 or 3 studies will direct the future in the treatment of patients harvesting these complex genomic alterations.

Upfront combinations of osimertinib with chemotherapy or other targeted agents might also result in the delay of the emergence of resistance, but still, we need to pursue curative strategies in this scenario.

To date, there are no available data on mechanisms of acquired resistance to osimertinib in the adjuvant setting; however, the increasing use of osimertinib in this early disease setting will probably lead to changes in the profile of resistance.

The technological advances, along with the creative designs of clinical trials in these infrequent subpopulations, will be key to improving the outcomes of patients with EGFR mutant NSCLC.

## Figures and Tables

**Figure 1 cancers-14-01931-f001:**
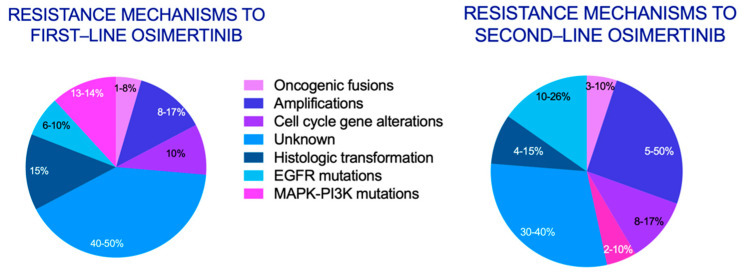
Resistance mechanisms to first- and second-line osimertinib. Adapted and reprinted from Leonetti et al. [16], Copyright © 2019, under exclusive license to Cancer Research UK. Cell cycle gene alterations (CCND1amp, CCND2amp, CCNE1amp, CDK6amp, CDKN2A E2fs), MAPK-PI3K mutations (*BRAF V600E*, *PIK3CA*, *KRAS*), oncogenic fusions (*FGFR*, *NTRK*, *RET*, *ALK*, *BRAF*), amplifications (*MET*, *HER2*, *PIK3CA*), EGFR mutations (C797X, L792X, G796X, exon20 insertions, etc.), histologic transformations (SCLC and SCC).

**Table 1 cancers-14-01931-t001:** Mechanism of resistance to osimertinib and proposed therapeutic options.

Mechanism of Resistance	Therapeutic Strategies	References
C797X	Gefitinib, erlotinib	[17]
	Osimertinib + erlotinib	[18]
	Brigatinib + cetuximab	[19]
	Patritumab deruxtecan	[20]
	EAI045	[21,22,23]
	JBJ-04-125-02	[24]
	CH7233163	[25]
	BLU-945	[26,27]
Small cell transformation	Platinum-etoposide	[28,29]
Squamous cell transformation	Histology based approach	[30]
*MET* amplification	Osimertinib + savolitinib	[31]
	Lazertinib + amivantamab	[32]
	Tepotinib + osimertinib	[33]
	Osimertinib + crizotinib	[34,35]
	Patritumab deruxtecan	[20]
*HER2* alterations	Patritumab deruxtecan	[20]
	Osimertinib + lapatinib *	[36]
	Osimertinib + neratinib *	[37]
	Osimertinib + T-DXd *	[38]
*RET* alterations	Osimertinib + selpercatinib	[39]
*BRAF* alterations	Osimertinib + dabrafenib + trametinib	[40]
	Osimertinib + vemurafenib	[41]
	Osimertinib + selumetinib or trametinib	[42]
*RAS*	Osimertinib + selumetinib or Aurora kinase b inhibitor	[43]
	Osimertinib + sotorasib *	[44]
	Osimertinib + adagrasib *	[45]
*PIK3*	EGFR TKIs and everolimus	[46,47]
	Osimertinib + alpelisib	[48]
Cell cycle gene alterations	Osimertinib + palbociclib	[49]
	Osimertinib + abemaciclib	[50]
AXL overexpression	Enapotamab vedotin	[51,52]
IGF-1 receptor activation	Osimertinib + linsitinib	[53]
Non-specific alterations	Datopotamab deruxtecan	[54]

* These proposed therapeutic mechanisms to overcome acquired resistance to osimertinib are yet to be tested. PI3K: Phosphoinositide 3-kinase, IGF-1: Insulin-like Growth Factor-1.

**Table 2 cancers-14-01931-t002:** Cell lines used to study resistance to osimertinib.

Induced Resistance Mechanism	Cell Lines	Mechanism of Induction	References
del19/T790M/C797S and L858R/T790M/C797S	NIH3T3 cells (immortalized mouse embryonic fibroblast cell line)	Transduction with lentiviruses	[25]
del19/L858R +/− T790M	Ba/F3 cells (a murine, IL-3 dependent, hematopoietic cell line)	Transduction with retroviral JP1540 or lentiviral JP1698 vectors	[55]
*MET* amplification	HCC827 cells (EGFR del19)	Exposure to osimertinib through a stepwise escalation process	[56]
HER2 exon 16 skipping	HEK293 cells (human embryonic kidney cell line and H1975 (T790M/L858R)	Plasmid transfection	[57]
*BRAF* G469A	PC9 cells (*EGFR* del 19)	Exposure to osimertinib through a stepwise escalation process	[42]
RAS alterations	PC9 cells	Exposure to osimertinib through a stepwise escalation process and a single concentration of osimertinib	[43]
Cell cycle gene alterations	H1975 cells (*EGFR* L858R/T790M)	Exposure to osimertinib through a stepwise escalation process	[49]
AXL overexpression	HCC827 cells (*EGFR* del19), PC9, H1975, and HCC4006 cells (*EGFR* del19)	Exposure to osimertinib through a stepwise escalation process a single concentration of osimertinib	[58,59,60]
Activation of IGF-1 receptor	PC9 cells	Exposure to gefitinib, developing resistance through the T790M, subsequently culture with stepwise escalation with osimertinib	[53]
	H1975 cells	Exposure to osimertinib using a high-concentration method	[53]
EMT	H1975/AR cells (gefitinib resistant)	Exposure to osimertinib through a stepwise escalation process	[61]
Other mechanisms: Src-AKT pathway and EGFR wild-type amplification	PC9 and H1975 cells	Exposure to osimertinib through a stepwise escalation process	[62]

## Data Availability

The data presented in this study are available in the article.

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
