# Peer review of "Acquired Mechanisms of Resistance to Osimertinib—The Next Challenge"

_cancers, 2022, doi:10.3390/cancers14081931_

Round 1
Reviewer 1 Report
The authors have addressed all the concerns raised and the manuscript has been modified appropriately. The revised manuscript is significantly improved and more readable. I have no additional concerns or comments.
Author Response
The authors have addressed all the concerns raised and the manuscript has been modified appropriately. The revised manuscript is significantly improved and more readable. I have no additional concerns or comments.
Response: Thank you very much for your observations.
Reviewer 2 Report
The manuscript has been revised well. I think this manuscript will be acceptable after some corrections have been done.
In line 79, authors should change C979X into C797X.
Author Response
The manuscript has been revised well. I think this manuscript will be acceptable after some corrections have been done.
In line 79, authors should change C979X into C797X.
Response: we changed it to C797X, thank you very much for your observations.
This manuscript is a resubmission of an earlier submission. The following is a list of the peer review reports and author responses from that submission.
Round 1
Reviewer 1 Report
This is a timely and useful review article on the mechanisms driving Osimertinib resistance in EGFR-mutant NSCLCs. The most commons mechanisms are described in a simple and direct fashion. There are several recent reviews on this topic and efforts should me made to make this review stand out in the crowd. A couple of suggestions that will make this review more useful to the readers, especially those who are not clinicians:
It might be good to expand EGFR the first time it is mentioned; one or two sentences describing the discovery/prevalence of EGFR mutations in NSCLC, especially in different races, as well as the earlier drugs used to combat it would be helpful in setting the stage. Further, it might be helpful to include 2-3 sentences, to highlight the commonalities in the resistance mechanisms against the first-generation inhibitors and Osimertinib.
One additional suggestion that would make this review more useful to laboratory-based scientists, including Clinician Scientists, is to include a Table that lists common Osimertinib/EGFR inhibitor resistant cell lines, with their mechanism of resistance. For example, it would be helpful for researchers to conduct initial experiments on cell lines that show secondary EGFR mutation (like H1975), SCLC conversion, Met and Her2 amplification, EMT etc., especially if it has been reported for Osimertinib resistance. Such information is widely available for resistance against first-generation inhibitors, and expanding this to Osimertinib resistance would be useful.
Some minor corrections/changes suggested are:
Line 86: Indicate the target of Brigatinib
Line 90:it would be good to show in parenthesis what ORR, DCR and mPFS stand for; especially DCR, which is used more sparingly in basic science literature.
Line 105: correct `aNSCLC’
Reviewer 2 Report
In this review paper, the authors described acquired resistance mechanisms to osimertinib and also summarized potential treatment strategies to overcome it. Comments from the reviewer are summarized below.
- Because mechanisms of resistance to osimertinib (including the frequencies of them) are different in 1st line and later line setting (especially for 2.2. Less common EGFR-dependent alterations). Therefore, the reviewer thinks the authors should describe the resistance mechanisms (and further treatment options) separately.
- The reviewer also feels that the manuscript is not well organized. For example, in Page 3, the authors described the efficacy of brigatinib plus cetuximab combination for cis-T790M/C797S, and then, described HER3-targeting antibody drug conjugate (not only for cis-T790M/C797S), 4th generation TKI, and front-line osimertinib + gefitinib combo.
- Many in vitro studies also reported potential mechanisms for osimertinib resistance. These data are not included in this manuscript.
- Resistance mechanisms should be heterogeneous in some patients. This would be a limitation to suggest resistance mechanism-based treatment (Table 1).
- The information should be added how the authors found publications for this review paper (PubMed? keywords? etc...).
Reviewer 3 Report
In this manuscript, the authors summarized acquired mechanisms of resistance to osimertinib. While this result extends our understanding of the acquired mechanisms of resistance to osimertinib, there is not entirely novel. Moreover, several points as indicated below need to be addressed by authors to improve the quality of the article.
Major points
- There are a lot of papers regarding the acquired mechanisms of resistance to osimertinib. This paper is not novel.
- There is not the mechanism of AXL (Taniguchi H et al. AXL confers intrinsic resistance to osimertinib and advances the emergence of tolerant cells. Nat. Commun. 2019;10:259. doi: 10.1038/s41467-018-08074-0) and IGF1R (Manabe T et al. IGF2 Autocrine-Mediated IGF1R Activation Is a Clinically Relevant Mechanism of Osimertinib Resistance in Lung Cancer. Mol. Cancer. Res. 2020;18:549–559. doi: 10.1158/1541-7786.MCR-19-0956) and so on. Moreover, Ramalingam et al have reported SPTBN1-ALK from FLAURA clinical trial.
Minor points
- English language and style are spell check required. I recommend this manuscript gets some English proofreading.
- Authors should add more information in Table 1.
- Authors should unify the font and the notation (HER2, Her2 and so on).
- The resolution of figures is low.